# One model fits all: Combining inference and simulation of gene regulatory networks

**Elias Ventre**[1,2,3☯¤a], **Ulysse Herbach**[4☯], **Thibault Espinasse**[2,3], **Gérard Benoit**[1¤b], **Olivier Gandrillon**[1,2]*

**1** Laboratoire de Biologie et Modélisation de la Cellule, École Normale Supérieure de Lyon, CNRS, UMR 5239, Inserm, U1293, Université Claude Bernard Lyon 1, Lyon, France, **2** Inria Center Grenoble Rhône-Alpes, Équipe Dracula, Villeurbanne, France, **3** Univ Lyon, Université Claude Bernard Lyon 1, CNRS UMR 5208, Institut Camille Jordan, Villeurbanne, France, **4** Université de Lorraine, CNRS, Inria, IECL, Nancy, France

☯ These authors contributed equally to this work.
¤a Current address: The University of British Columbia, Vancouver, Canada
¤b Current address: Université de Rennes 1, IGDR - CNRS UMR 6290 - Inserm 1305, Rennes, France
* olivier.gandrillon@ens-lyon.fr

**Data Availability Statement:** CARDAMOM is available at https://github.com/eliasventre/cardamom along with the code to generate all the

## Abstract

The rise of single-cell data highlights the need for a nondeterministic view of gene expression, while offering new opportunities regarding gene regulatory network inference. We recently introduced two strategies that specifically exploit time-course data, where single-cell profiling is performed after a stimulus: HARISSA, a mechanistic network model with a highly efficient simulation procedure, and CARDAMOM, a scalable inference method seen as model calibration. Here, we combine the two approaches and show that the same model driven by transcriptional bursting can be used simultaneously as an inference tool, to reconstruct biologically relevant networks, and as a simulation tool, to generate realistic transcriptional profiles emerging from gene interactions. We verify that CARDAMOM quantitatively reconstructs causal links when the data is simulated from HARISSA, and demonstrate its performance on experimental data collected on *in vitro* differentiating mouse embryonic stem cells. Overall, this integrated strategy largely overcomes the limitations of disconnected inference and simulation.

## Author summary

Gene regulatory network (GRN) inference is an old problem, to which single-cell data has recently offered new challenges and breakthrough potential. Many GRN inference methods based on single-cell transcriptomic data have been developed over the last few years, while GRN simulation tools have also been proposed for generating synthetic datasets with realistic features. However, except for benchmarking purposes, these two fields remain largely disconnected. In this work, building on a combination of two methods we recently described, we show that a particular GRN model can be used simultaneously as an inference tool, to reconstruct a biologically relevant network from time-course single-cell gene expression data, and as a simulation tool, to generate realistic transcriptional

results in this paper. HARISSA is available at
https://github.com/ulysseherbach/harissa.

**Funding:** This work was supported by funding
from French agency ANR (SingleStatOmics; ANR-
18-CE45-0023-03) to OG. The funders had no role
in study design, data collection and analysis,
decision to publish, or preparation of the
manuscript.

**Competing interests:** The authors have declared
that no competing interests exist.

profiles in a non-trivial way through gene interactions. This integrated strategy demonstrates the benefits of using the same executable model for both simulation and inference.

## Introduction

Cell decision making as a response to exogenous or endogenous stimuli (e.g., differentiation, proliferation, cell death or biological activity modulation) is often supported by time-dependent modulation of gene expression upon stimulation. Understanding how and why gene expression changes as a function of time in response to specific stimuli is therefore critical to understand the underlying biological processes.

The "how" question can now be approached using single-cell-based technologies, offering an unprecedented resolution and a much finer view than population-based measures [1, 2]. The "why" question relates to the functioning of an underlying gene regulatory network (GRN) which describes interactions between genes through their expression products. GRNs are thus a central notion for understanding and predicting cellular behavior, but their construction from literature is a very laborious task, sometimes even impossible due to the lack of knowledge.

Reconstructing most-likely GRNs from transcriptomic datasets has therefore become a major goal in systems biology [3] but is also notoriously difficult, especially in the case of single-cell transcriptomic data. Indeed, the bursty synthesis of mRNAs, now clearly evidenced [4, 5], gives rise to highly variable and non-Gaussian expression data [1, 6], and current GRN inference methods employ a wide range of statistical and modeling tools [7]. Methods based on a specific dynamical model, called here GRN models, have the great advantage of providing biological interpretability, since each inferred interaction between genes can be understood in terms of model behavior. Moreover, such approach generally provides interactions with their direction and intensity, which is not the case for most purely statistical methods.

In this article, we make a distinction between mechanistic GRN models (e.g., built on the biological understanding of differentiation mechanisms) for which cell behavior appears as an emergent property of gene interactions, and phenomenological models, for which the expected outcome is directly prescribed by some dedicated parameters. In this case, although such parameters can still have a biological meaning, the cellular behavior is not biologically emerging but rather 'hard-coded' by the model. As developed afterwards, many GRN models fall in between: some aspects of gene expression patterns are then hard-coded instead of emerging from interactions between genes, and gene expression stochasticity is often assumed to be driven by Gaussian white noise only, requiring ad hoc additional noise to match the data.

Moreover, the results of a method based on a mechanistic model can only be considered relevant if the model is able to correctly reproduce single-cell datasets. For instance, it is now widely accepted that the transcriptional bursting phenomenon is associated to specific patterns of gene expression products [8, 9], making continuous single-cell data close to Gamma distributions [10] and discrete data close to negative binomial distributions [11], the latter being themselves mathematically equivalent to Poisson distributions with Gamma-distributed random parameters. Thus, executable network models should at least be able to generate these patterns in their marginal distributions. In any case, the use of a mechanistic model-based method requires prior strong evidence that the underlying model is relevant for simulating realistic single-cell transcriptomic datasets.

We recently developed several methods for inferring GRNs from single-cell data based on a particular mechanistic network model, defined as a 'multi-agent' generalization of the well-

known two-state stochastic model of gene expression [8] where genes are now being described by interacting two-state models [6]. These methods are well suited for single-cell RNA-seq (scRNA-seq) time-course data, each dataset being considered as a partial observation of the model at a certain time. Crucially, they do not require the observation of cell trajectories, whose inference is a problem in itself [12, 13], but only that the cells sampled at each timepoint are driven by the same dynamical process, i.e., resulting from the same GRN. Our first proposal was called WASABI [14], which uses a divide-and-conquer approach where the problem of GRN inference is solved one gene at a time. Although able to propose relevant GRNs, this approach suffered from two drawbacks: it required days of computation for a GRN with 50 genes, and proposed a potentially long list of candidate networks. We therefore developed two other methods: HARISSA [6], a GRN simulation algorithm based on the mechanistic model together with a proof-of-concept inference method derived from likelihood maximization, and CARDAMOM [15], a simplified and scalable alternative for the GRN inference part that crucially exploits the notions of landscape and metastability.

In this work, we sought to investigate the benefits of using this model as an integrated tool for both GRN inference and data simulation. We therefore assessed its ability to allow for efficient network reconstruction from time-course scRNA-seq data, while accurately reproducing the dataset main features from the functioning of the inferred network. Note that to the best of our knowledge, this is not performed by existing GRN-based simulation tools, which are generally based on more phenomenological than mechanistic models, with at least some important aspects of gene expression patterns, such as transitions between cell types [16] or gene expression variability [16, 17], being hard-coded instead of arising from biological mechanisms.

After introducing the setup of our benchmark made from *in silico* datasets generated with the mechanistic model, we first evaluate the performances of HARISSA and CARDAMOM together with four state-of-the-art GRN inference algorithms: GENIE3 [18], PIDC [19], SINCERITIES [20] and SCRIBE [21]. We study the limits of the different categories of inference methods in the case of transcriptional bursting, and verify that the two model-based methods perform better than the others on these datasets. CARDAMOM appears as the best performing algorithm during this benchmark step, which only considers network structures. Importantly, the output of this algorithm is not only a matrix of interaction scores, but also a set of quantitative parameters that can be plugged into the GRN model for simulations.

In a second step, we use CARDAMOM to calibrate the model with a real time-stamped scRNA-seq dataset of differentiating mouse embryonic stem (ES) cells [22]. We demonstrate the ability of the model to reproduce the global features of real time-course transcriptomic profiles. We also show that most of the inferred interactions are indeed supported by biological evidence such as ChIP-seq experiments, although this evidence was not used during the inference process. Altogether, these results establish the ability of an executable network model not only to simulate realistic single-cell datasets, but also to provide an effective reverse-engineering algorithm capable of reproducing the main gene expression patterns of an experimental dataset as emergent properties of the underlying GRN.

## Results

### HARISSA simulates single-cell datasets from a mechanistic GRN model

We first wanted to benchmark the ability of the different inference algorithms to reconstruct correct network structures from *in silico* generated datasets, i.e., when the ground truth is known. For this, we used the simulation module of HARISSA [6], which generates trajectories of a mechanistic model describing gene expression dynamics (both mRNA and the

corresponding proteins) within a single cell, these dynamics being influenced by an underlying GRN and driven by transcriptional bursting (see Simulation of the inferred network reproduces the original dataset and S1 Fig). As shown in previous work, this model is indeed able to generate scRNA-seq datasets with realistic marginal distributions [23].

We simulated nine datasets corresponding to different network structures (Fig 1): a network of 4 genes with a branching structure and inhibition feedback loop (FN4); a network of 5 genes with a cycling structure (CN5); a network of 8 genes with multiple branching structure and feedback loops (FN8); a network of 8 genes with branching trajectories (BN8); networks with a tree structure of 5, 10, 20, 50, and 100 genes (Trees). These networks represent the main types of network structures that have been used for benchmarking GRN inference algorithms [17]. Overall, the objective was to reproduce time-course experiments in which single-cell profiling is performed after a given stimulus, typically a change of medium [22, 24, 25]. This stimulus was therefore taken into account in all the simulations, in the form of a virtual gene defined as being inactive before the beginning of the experiment and fully activated afterwards.

For each network structure (Fig 1A), the transcriptional bursting model implies that typical single-cell trajectories do not follow a diffusion-like process (at least in the space of mRNA levels), and differ strongly from the more usual and intuitive population-average trajectory (Fig 1B). The practical datasets were obtained by sampling independent cells at a specific sequence of timepoints, therefore not keeping the real cell trajectories but rather considering different cells at each timepoint, forming time-stamped snapshots (Fig 1C). Interestingly, both feedback networks (FN4 and especially FN8) produce a recognizable "differentiation trajectory" across the UMAP space with a clear temporal order of cells. Due to the stochastic nature of cell trajectories generated by the mechanistic model, branching trajectories in snapshots only appear in specific cases, generally when a toggle-switch is dominating the GRN structure and then generates distinct branches in the UMAP representation (see BN8 in Fig 1).

As mentioned previously, HARISSA consists of two modules for performing respectively simulation and inference. Whereas the original inference module of HARISSA was limited to a few genes [6], it recently integrated an effective CARDAMOM-inspired simplification [23] that allows to infer networks with a much larger number of genes. We therefore also benchmarked this method along with the others.

## CARDAMOM quantitatively reconstructs causal GRN links

We then inferred GRN structures from the *in silico* generated datasets using the six algorithms presented in the Simulation of the inferred network reproduces the original dataset section (HARISSA, CARDAMOM, GENIE3, PIDC, SINCERITIES, and SCRIBE). Note that neither GENIE3 nor PIDC are able to use the temporal information (except for the stimulus state information, which they are also provided with), giving them a disadvantage compared to the other algorithms. They were nevertheless used in the benchmark as they are considered to be among the best algorithms for single-cell data, and given that very few algorithms are specifically adapted to time-stamped datasets. Indeed, most methods are limited to static data, and those that are not (such as SCRIBE) require temporally-ordered cell trajectories instead of independent snapshots, thus requiring a pre-processing step that can itself be subject to errors. Moreover, it was not known how they would fare in a time-course setting with transcriptional bursting, which was an interesting question per se.

We also emphasize that among these algorithms, only CARDAMOM and HARISSA have the significant advantage of providing biological interpretability, thanks to the mechanistic model on which they are based: here the network parameters are not mere interaction scores, but quantitative parameters that can be plugged into the model for simulations. Besides, the

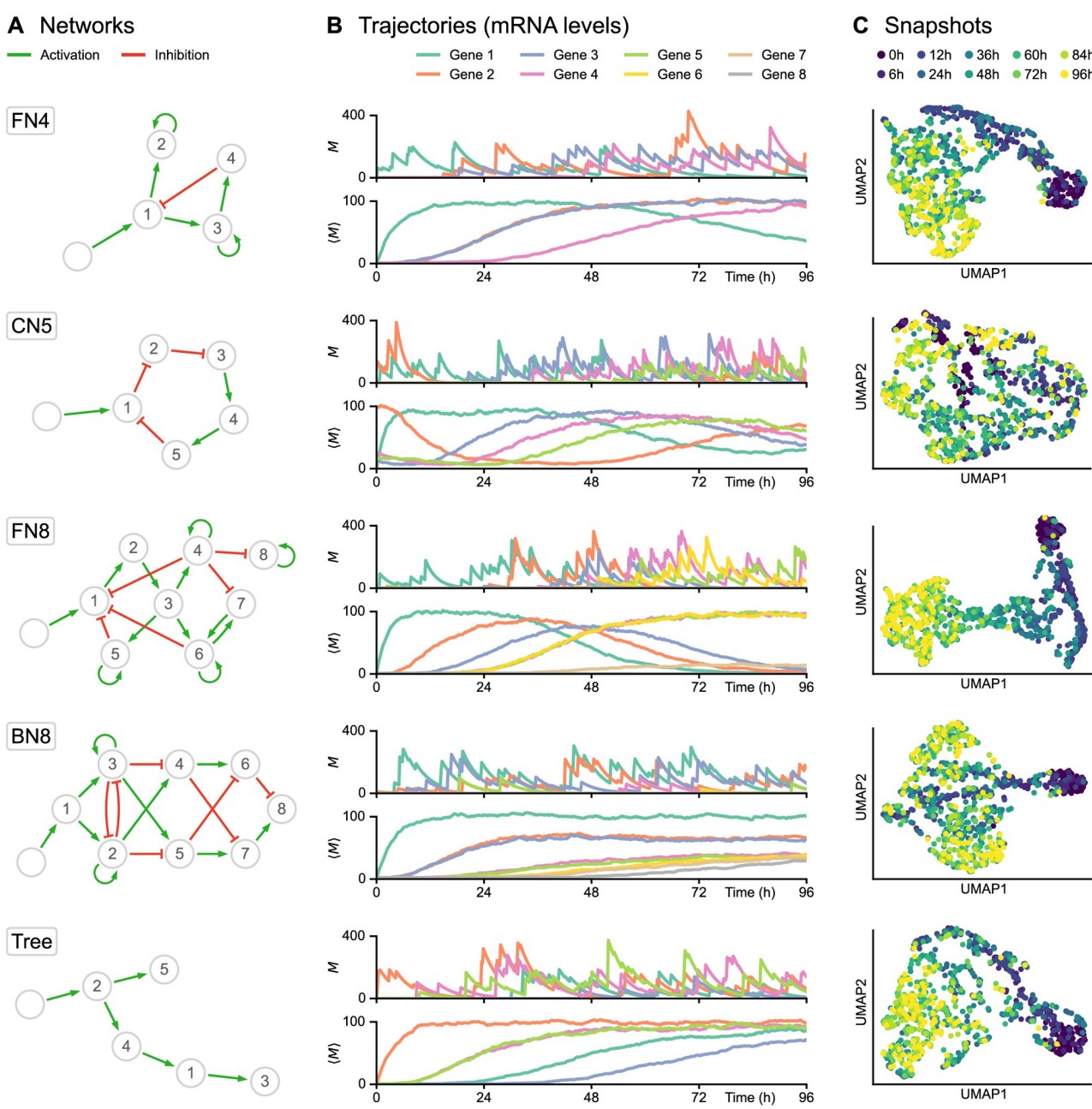

**Fig 1. Single-cell data simulation using HARISSA. (A)** Networks used for subsequent tests, including feedback loop networks (FN), a cycling network (CN) and a branching network (BN). Genes and stimulus are represented by numbered nodes and an empty node, respectively, while green arrows indicate activation and red blunt arrows indicate inhibition. **(B)** Corresponding trajectories, defined as time-dependent mRNA levels (in copies per cell). For each network, the first plot shows one example of single-cell trajectory $M$ while the second plot shows the population average $\langle M \rangle$ from 1000 cells. The transcriptional bursting model underlying HARISSA implies that every single-cell trajectory differs strongly from the more usual population average. **(C)** Two-dimensional UMAP representations of corresponding single-cell snapshots, defined as mRNA levels sampled at 10 timepoints in different cells from 0h to 96h, with 100 cells per timepoint. Such snapshots are called *time-stamped data* in the text and are fundamentally different from single-cell trajectories, which are currently not available experimentally.

main objective is really to reverse engineer such a model: from this perspective, despite the obvious advantage of CARDAMOM and HARISSA being built on the same mathematical framework as the one used to generate the data, even similar performances compared to the other algorithms would be satisfying.

Inference was performed on ten independent datasets for each network, and the results were merged into the area under the precision-recall curve (AUPR) which measures the quality of the inferred GRN structure. We also compared the inferred GRNs with a naive method consisting in assigning to each edge of the network the value given by the Pearson correlation coefficient between the corresponding genes (abbreviated as PEARSON): this comparison with Pearson coefficients makes it possible to verify, when the algorithms show good performances, that these are not only due to highly correlated data which are thus not difficult to analyze. The results are presented in Fig 2A and 2B for the first five algorithms. We present the results for SCRIBE separately in Fig 2C because this algorithm requires temporally or pseudo-temporally ordered trajectories, and the results then depend on the pre-processing that is applied on the time-stamped data.

CARDAMOM and HARISSA appeared to outperform the other algorithms for most of these datasets. In particular, in terms of directed interactions, these two methods always clearly performed better than the others. The undirected networks for which GENIE3 and PIDC have similar performances (CN5 and BN8) correspond to cases where the Pearson correlation method is also accurate.

Also, if GENIE3 and PIDC represent an improvement over the Pearson correlation method, they seem to perform poorly when the correlation between genes is not sufficient to infer a reliable GRN. More precisely, we observe that GENIE3 and PIDC are accurate for tree-like networks (Trees), even with bifurcating trajectories (BN8) and cycling (CN5), which was not the case in [17]. On the contrary, SINCERITIES performs very poorly for these type of networks, but seems however competitive for networks with feedback loops (FN4 and FN8) where GENIE3 and PIDC have lower performances. These networks are more difficult to reconstruct. Indeed, as visible in Fig 1, the population-average trajectories of some genes are completely similar. Some genes also have the same marginal distribution of mRNA levels: for example in the network FN4, gene 2 and gene 3 have the same input (gene 1), so their marginal distributions evolve similarly at each timepoint. Then SINCERITIES, which bases the inference procedure on the approximate distribution for each gene, fails to make this subtle distinction, illustrating the improvement that is typically expected from HARISSA and CARDAMOM. On all the networks, GENIE3 fails to infer reliably the direction of the interaction, i.e., to distinguish the interaction $i \rightarrow j$ from the interaction $j \rightarrow i$. On the contrary, because of their mechanistic assumptions, CARDAMOM, HARISSA and SINCERITIES have always quite similar results for directed and undirected inference. Finally, we observed that CARDAMOM outperforms HARISSA on most of the networks.

Regarding SCRIBE, which is a trajectory-based method, we tested its performances in three scenarios (Fig 2C):

1. When we have access to real trajectories (*Real traj.*): each cell at each timepoint is being associated to a real ancestor at the previous timepoint and a real descendant at the following one. Such knowledge can of course only be accessed with *in silico* generated datasets or *in vitro* for a very limited number of genes by using live-cell imaging of short-lived transcriptional reporters [26];

2. When the dataset is the same as for the other methods (i.e., no access to real trajectories), and each cell at each timepoint is associated to a pseudo-ancestor at the previous timepoint and a pseudo-descendant at the following one, using the Waddington-OT method described in [27] (*Coupling*);

3. When the dataset is the same as for the other methods (i.e., no access to real trajectories), and the algorithm SLINGSHOT [28] is used for reconstructing a pseudo-temporally ordered trajectory (*Pseudotime*).

 

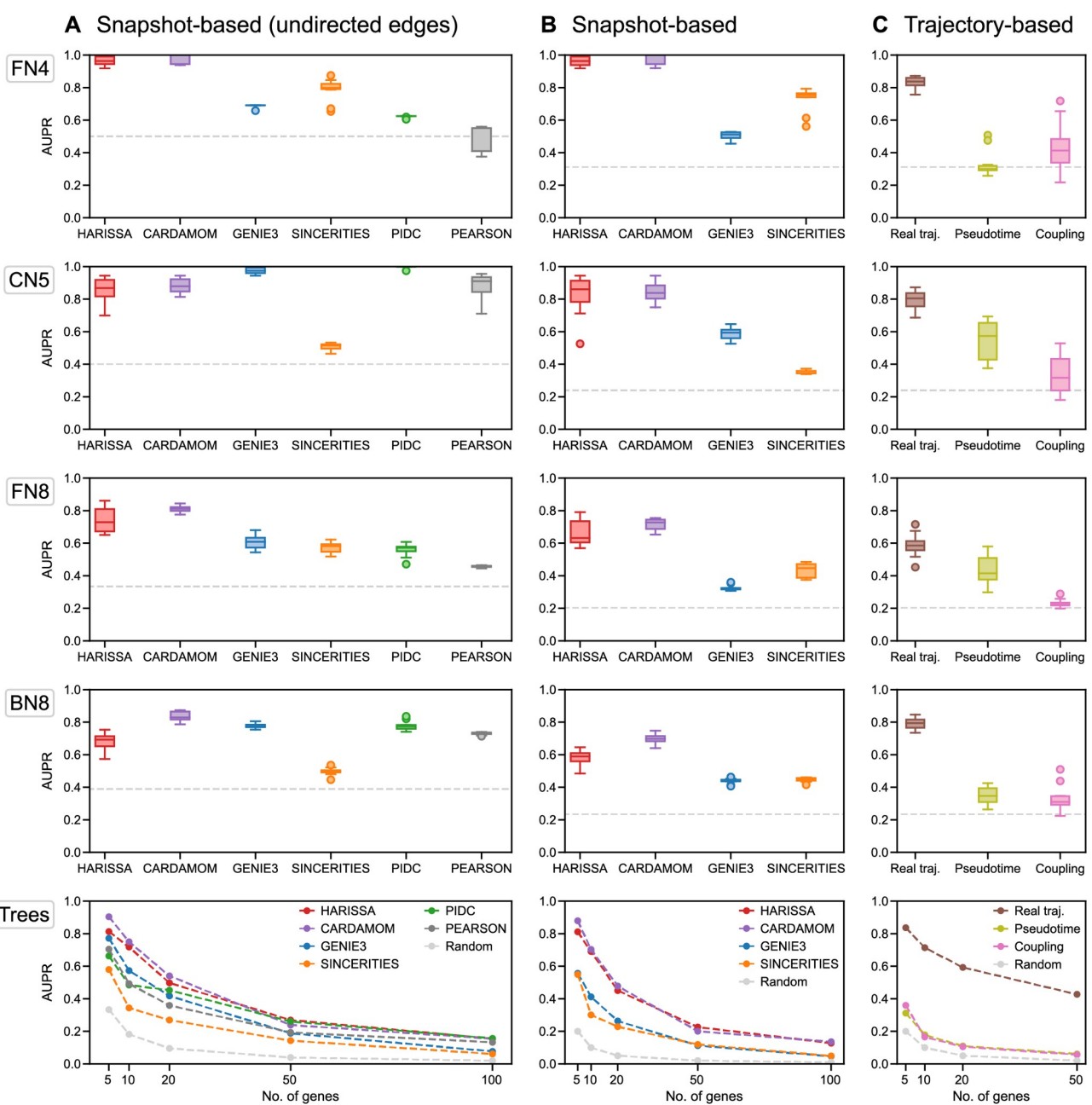

**Fig 2. Benchmark of inference methods for five different network structures.** For each network, inference is performed on ten independently simulated datasets, each dataset containing the same 10 timepoints with 100 cells per timepoint (1000 cells sampled per dataset). The performance on each dataset is then measured as the area under the precision-recall curve (AUPR), based on the unsigned inferred weights of edges. Finally, the performance of each method is summarized as a box plot of the corresponding AUPR values, or the average AUPR value for the tree-structure activation networks (*Trees*). For each plot, the dashed gray line indicates the average performance of the random estimator (assigning to each edge a weight 0 or 1 with 0.5 probability). For the *Trees* networks, each dataset corresponds to a random tree structure of fixed size (5, 10, 20, 50, and 100 genes) sampled from the uniform distribution over trees of this size. **(A)** Performance of all methods when considering only undirected interactions. **(B)** Performance of the methods able to infer directed interactions. **(C)** Performance of the SCRIBE inference method for the same networks, in three conditions: when one has access to real single-cell trajectories (in brown), when pseudo-trajectories are reconstructed from time-stamped data using a coupling method similar to Waddington-OT (in pink), and when a single pseudo-trajectory is reconstructed using the pseudotime algorithm SLINGSHOT (in light green). For the last two conditions, the datasets used are therefore the same as those used for the other methods.

 

We observed that SCRIBE performs well in scenario 1, but poorly in scenarios 2 and 3, at least on the tested networks (Fig 2C). These poor performances are due to the loss of temporal coupling between measurements of genes that interact. They suggest that neither optimal coupling nor pseudotime reconstruction are sufficiently efficient for GRN inference in case of transcriptional bursting. Concerning the optimal coupling method, we notice that this might be due to the "movement by diffusion" assumption on which the Waddington-OT method is built, which does not take into account the constraints on the trajectories imposed by the GRN.

When computing the average runtime of each algorithm on the tree-like networks, we observed that except for SCRIBE, all algorithms are suitable for inferring GRN with a realistic number of genes (see S1 Table). Thus, due to this computational limit and its poor performances when using time-stamped data, we did not consider SCRIBE for further analysis.

We then investigated the limit performances with respect to the number of cells and/or timepoints. We observed that the performances of the first five algorithms decrease for the tree-like networks when the number of genes increases (Fig 2). This can be due to three main factors:

1. A sequence of timepoints too coarse in relation to the dynamics would directly lead to a lack of inference accuracy;

2. A sequence of timepoints which is too restricted may not allow to see interactions involving some genes that are regulated late in the process. For example, in Fig 2, we observe that the inference on the Trees networks is very poor for more than ten genes: it comes from the fact that some genes are never activated before 96h;

3. The number of cells at each timepoint can simply not be enough to infer a reliable GRN.

We therefore investigated the effects of these three factors on the accuracy of the algorithms by studying their performances in terms of AUPR for ten datasets generated from ten randomly-generated tree-like network of ten genes, when varying the number of cells at each timepoint (Fig 3A), the length of the interval for a fixed time gap between each timepoint (Fig 3B), and the density of the sequence of timepoints for an interval with fixed length (Fig 3C). As anticipated, all these conditions have an impact on the quality of the inference: augmenting their values tends to produce a better quality of inference. We also observed that the number of sampled cells seems less critical than the other factors, confirming that few cells at a sequence of timepoints which is dense and long-enough is preferable to many cells on a sequence of timepoints which is too coarse and/or too short. This should be kept in mind when designing single-cell transcriptomics experiments aiming at GRN inference.

Hence, both CARDAMOM and HARISSA, with a benefit for using CARDAMOM, allowed to efficiently reconstruct network structures by reverse engineering the generative model on which they are based. We then needed to test its ability to reproduce an experimental dataset from the literature after network inference.

## Application to a real dataset yields a biologically relevant network

As a test case, we used a time-stamped *in vitro* dataset from Semrau et al. [22] obtained by scRNA-seq of a retinoic acid (RA)-induced differentiation of mouse ES cells (see Simulation of the inferred network reproduces the original dataset). This well-characterized model system of *in vitro* differentiation recapitulates the transition from pluripotent embryonic stem cells towards two cellular lineages (ectoderm- and extraembryonic endoderm-like cells), all characterized by well-established molecular markers that were further used in GRN inference.

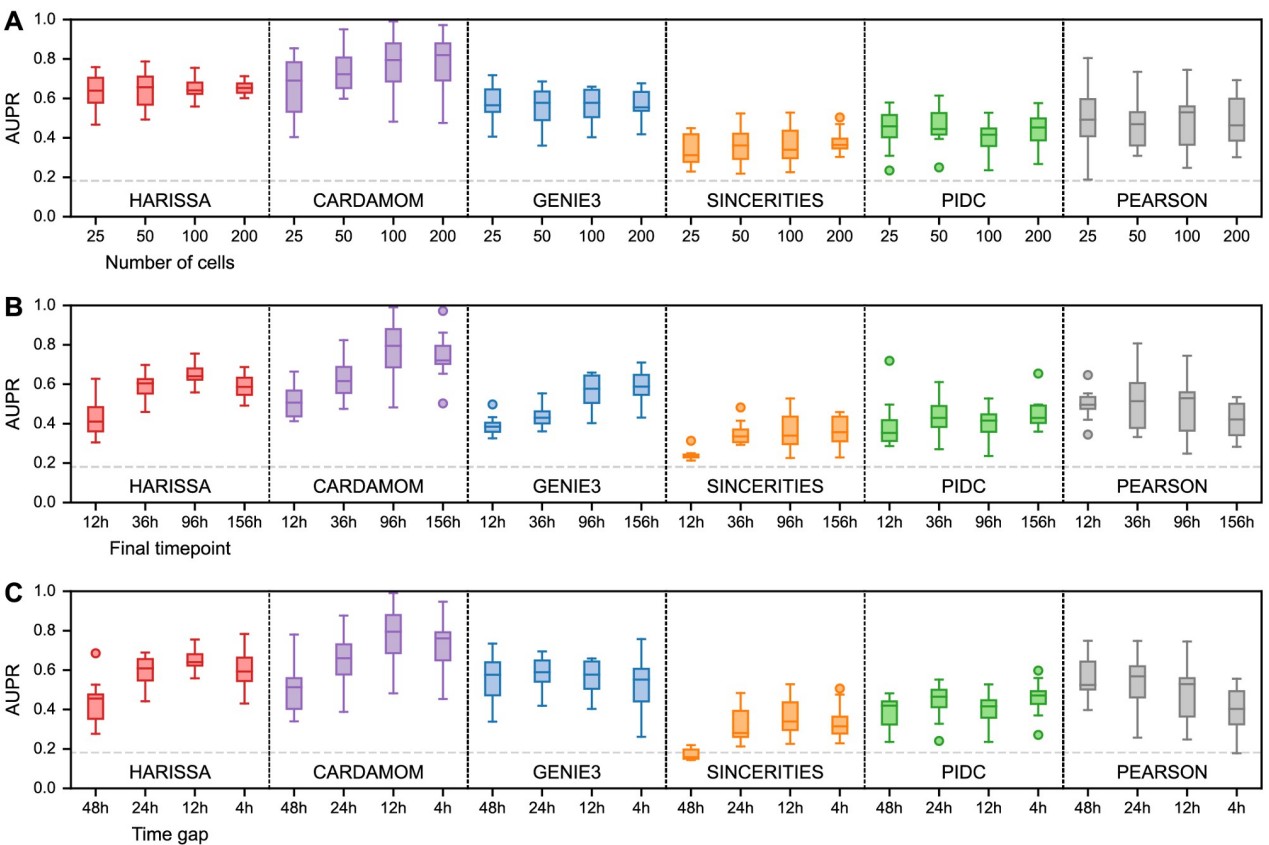

**Fig 3. Dependence of inference methods on data collection parameters.** For simplicity, only the case of undirected interactions is considered here and the datasets are restricted to 10-gene tree-structure networks (see Fig 2 for the general benchmark). Inference is performed for each method and condition on ten independently simulated datasets and summarized by box plots of AUPR values as in Fig 2. For each plot, the dashed gray line indicates the average performance of the random estimator (assigning to each edge a weight 0 or 1 with 0.5 probability). **(A)** Performance as a function of the number of cells per timepoint, while keeping the same timepoints. **(B)** Performance as a function of the length of the measurement period, while keeping the same gap between timepoints and the same total number of cells. **(C)** Performance as a function of the gap between timepoints, while keeping the same final timepoint and the same total number of cells.

In order to interpret the resulting GRN, we sought to assess whether the inferred interactions are supported by known biochemical evidence of physical interaction between regulators and regulated genes (Fig 4). For this, we annotated the inferred edges coming from genes encoding known transcription regulators (i.e., transcription factors and cofactors) included in the network and for which ChIP-seq data are currently available in ES or the closely related embryonic carcinoma (EC) cell system. Since the RA stimulus exerts its differentiating effect mainly through the members of the RA-activated nuclear receptors subfamily RAR (NR1B) that encompass 3 paralogs (i.e., RARα, RARβ and RARγ), the annotation of the interaction edges linking the stimulus and the regulated genes was based on the presence/absence of ChIP-seq peaks for any RAR paralogs at less than 10 kb upstream or downstream of the annotated transcription start site (TSS) in RA-stimulated ES or EC cells [29, 30]. Although arbitrary, the chosen distance between TSS and DNA binding site for the indicated transcription regulator is relatively conservative as transcriptional effect could be exerted from greater distance up to megabases [31] and the absence of supporting peak as defined should not be interpreted as a proof of absence of any direct modulating effect. Similarly, the edges supported by physical interactions data for Sox2 and Pou5f1, or Jarid2 were extracted from [32, 33].

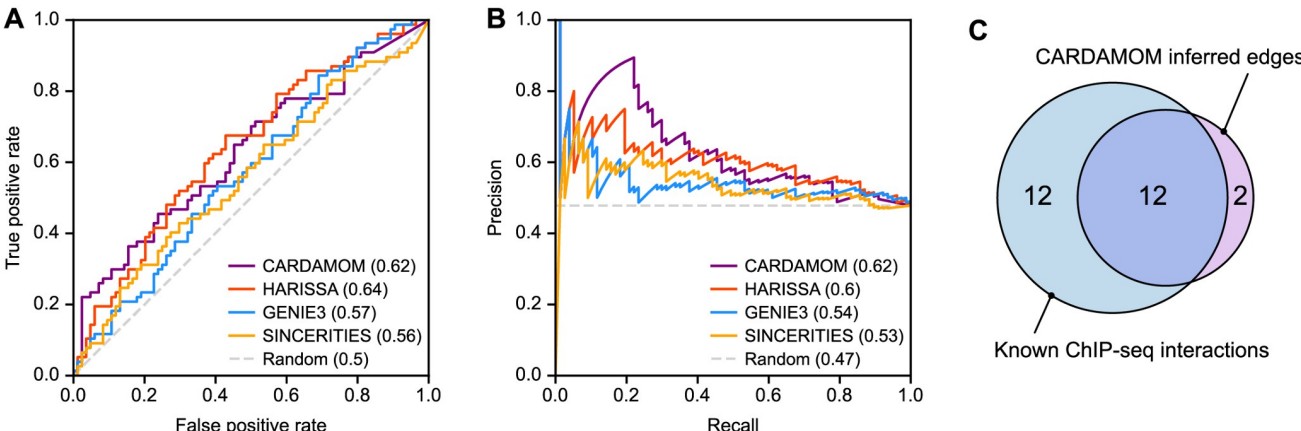

**Fig 4. Comparison between inference methods and physical interactions derived from ChIP-seq.** The four directed GRN inference methods were applied to the experimental dataset from [22] restricted to a panel of 41 marker genes identified by the authors, and a reference network was obtained independently from edges supported by ChIP-seq data. As we only have access to physical interactions involving the retinoic acid (RA) stimulus or genes Pou5f1, Sox2, and Jarid2, the comparison only considers the $4 \times 41$ related edges. **(A)** Receiver operating characteristic (ROC) curve and corresponding area under the curve (AUROC) for each inference method. **(B)** Precision-recall (PR) curve and corresponding area under the curve (AUPR) for each inference method. **(C)** Venn diagram showing the overlap, for interactions involving the RA stimulus, between directed edges predicted by CARDAMOM and known physical interactions identified by ChIP-seq analysis.

Using these known physical interactions as a ground-truth, we compared the receiver operating characteristic (ROC) and precision-recall (PR) curves related to the network structures inferred by the four algorithms (Fig 4A and 4B). We observed that, in accordance with the previous results, CARDAMOM and HARISSA appear as the top-ranked algorithms, displaying both a very close ability to infer known edges.

We then examined the structure of the network inferred by CARDAMOM (Fig 5). Importantly, in agreement with its differentiating effect in ES/EC cell systems, we observed that the RA stimulus is densely connected with genes involved in pluripotency maintenance as supported by multiple biological analyses [29, 30, 34] and to a lesser extent with gene nodes corresponding to genes associated with specific cell fates, this latter observation likely reflecting how the stimulus is modeled (see Simulation of the inferred network reproduces the original dataset). Notably, these last nodes also exhibit a relatively high interconnectivity (e.g., endodermal differentiation) as compared to intergroup connectivity. Although biologically interesting, these observations illustrate our previous conclusion and likely mirrors the unbalanced experimental design characterized by a dense sequence of timepoints during the early phase (0h to 36h) of the differentiation process analysed and a coarser sequence of timepoints in the mid (36h to 48h) and late (48h to 96h) phases of the process [22].

Most notably, the overwhelming majority (85%) of the inferred edges that involve the RA stimulus are supported by biochemical evidence (Fig 4C). Similarly, the edges inferred from Pou5f1, Sox2, and Jarid2 nodes are globally supported by physical interaction (2/3 for Pou5f1, 2/3 for Sox2, and 1/1 for Jarid2).

We also observed that some inferred edges are not supported by documented physical interactions, as expected for genes encoding proteins unable to directly interact with DNA. As an example of such node, Sparc (also known as Osteonectin) appears highly connected to genes associated with all four cell states despite its inability to directly interact with gene basal transcription machinery (i.e., RNA polymerase complex). However, the inferred edges are clearly in agreement with its documented role in promoting endodermal differentiation [35]. Additionally, unsupported inferred edges may mirror the lag time between the expression and

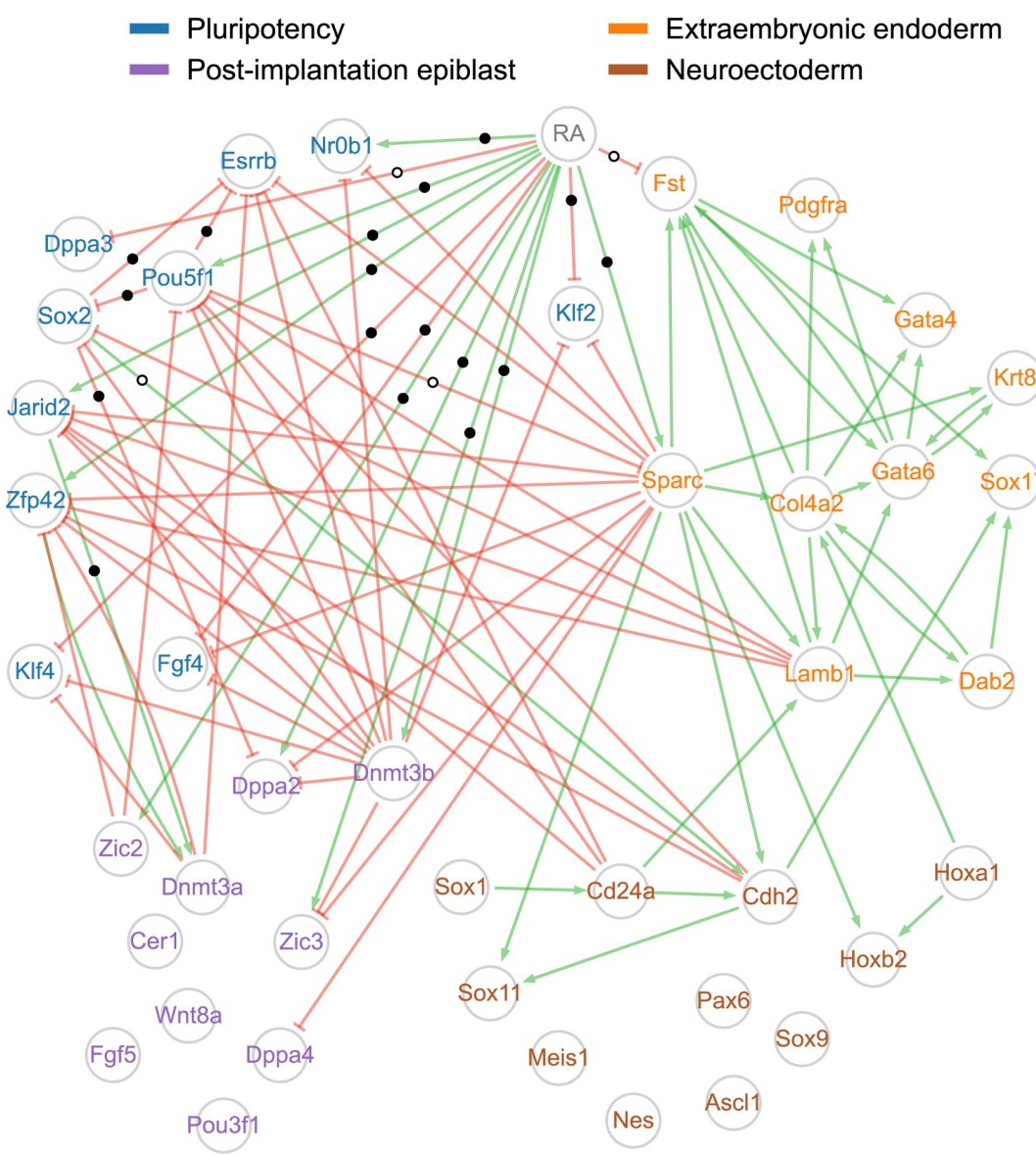

**Fig 5. Network inferred by CARDAMOM from a real time-stamped scRNA-seq dataset.** The CARDAMOM inference method was applied to the experimental dataset from [22] restricted to a panel of 41 marker genes identified by the authors. The network structure is obtained by keeping only the 5% strongest activations (green arrows) and inhibitions (red blunt arrows) acting on each gene. Genes are colored according to four groups related to different cell states (pluripotency, post-implantation epiblast, neuroectoderm, extraembryonic endoderm) following the proposed classification of [22]. Edges supported by a ChIP-seq interaction are marked with black dots (see main text for the definition of what is considered as an interaction) and edges that are not supported are marked with white dots: this concerns only the edges starting from the RA stimulus, Pou5f1, Sox2, and Jarid2. Edges for which we have no reliable information have no mark.

therefore the physical interaction between regulator and regulated genes and the observed transcriptional effect. By contrast with TFs that establish contact with the transcription machinery, modifying cofactors often catalyse deposition/erasure of epigenetic marks (e.g., acetylation/methylation of histones, DNA methylation) that will likely modulate transcription in a longer lasting manner. In this respect it is interesting to note that the Dnmt3b gene negatively interacts with many other genes in the network, which mirrors the fact that Dnmt3b is a

*de novo* DNA methyltransferase, and has an indirect effect on gene regulation through CpG methylation, a well documented epigenetic mark generally associated with gene expression silencing. Altogether, this illustrates that our GRN model does incorporate various epigenetic information or indirect effects and is not restricted to physical interactions between transcription factors and their target genes.

While most inferred edges involving genes that encode TFs appear to be supported by physical interactions, many physical interactions detected by ChIP-seq are missed by CARDAMOM (e.g., 50% for RA, see Fig 4C). This observation is however not necessarily the sign of a lack of accuracy of the inferred GRN, since the detection of a physical interaction is not *per se* the hallmark of a modulating effect on the transcription level of the target gene [30]. Additionally, some specific regulatory structures are notoriously difficult to infer as illustrated by the high failure rate (96%) in inferring edges from Jarid2, a component of a repressive complex expressed in the pluripotent state and directly involved in the silencing of differentiation-associated genes. Interestingly, the interaction between Jarid2 and most of its physical targets presented in our GRN were instead wrongly detected as an inhibitory effect of the regulated genes on their regulators. This is due to the fact that CARDAMOM works by going forward in time, and thus fails to capture an inhibition that has an effect at the beginning of the process and which can be detected only further: instead, it would be prone to interpret the increase of the repressed genes by the effect of other intermediate genes, and the decrease of the repressor by an inhibitory effect of the repressed genes. We discuss further such bias of the algorithm in the Simulation of the inferred network reproduces the original dataset section.

For a better understanding of the inferred GRN dynamics, we also examined a dynamical network representation, where each edge appears at the timepoint for which it was detected with the strongest intensity by the inference algorithm (Fig 6). Unsurprisingly, the RA stimulus is detected at the earliest timepoint of the response (6h) and then ceases to influence the signal, which propagates in waves through the network as we described in a previous study [14]. For example, we do clearly observe the late increase of interactions for genes involved in specifying the extraembryonic endoderm.

The network inferred by SINCERITIES is shown in S7 Fig, which can be compared with Fig 5. Although a detailed analysis of this network is left to the interested reader, we observe that while some important characteristics of the network structure are similar to the ones of CARDAMOM (RA stimulus highly connected with genes involved in pluripotency maintenance, high connectivity of Sparc and Pou5f1), the correspondence ratios with ChIP-seq data are not as good as for CARDAMOM (in agreement with Fig 4A and 4B) as SINCERITIES seems to infer more interactions at the cost of more errors.

## Simulation of the inferred network reproduces the original dataset

While inferring the GRN structure, CARDAMOM also inferred all the other parameters of the model, as described in Simulation of the inferred network reproduces the original dataset, except the mRNA degradation rates $d_{0,i}$ for each gene $i$. These parameters $d_{0,i}$ are not negligible as they scale the dynamics of the process. To address this problem, we used values from the literature that can be found in [36] (see Simulation of the inferred network reproduces the original dataset and S2 Table). Once the model has been calibrated, one can simulate an *in silico* dataset and sample the nine timepoints corresponding to the *in vitro* experiment. More precisely, we simulated two different datasets after calibrating the mechanistic model: one actually using the inferred network interactions, and one corresponding to the "null network" defined by removing the interactions (i.e., all genes individually calibrated with the same parameters but kept independent).

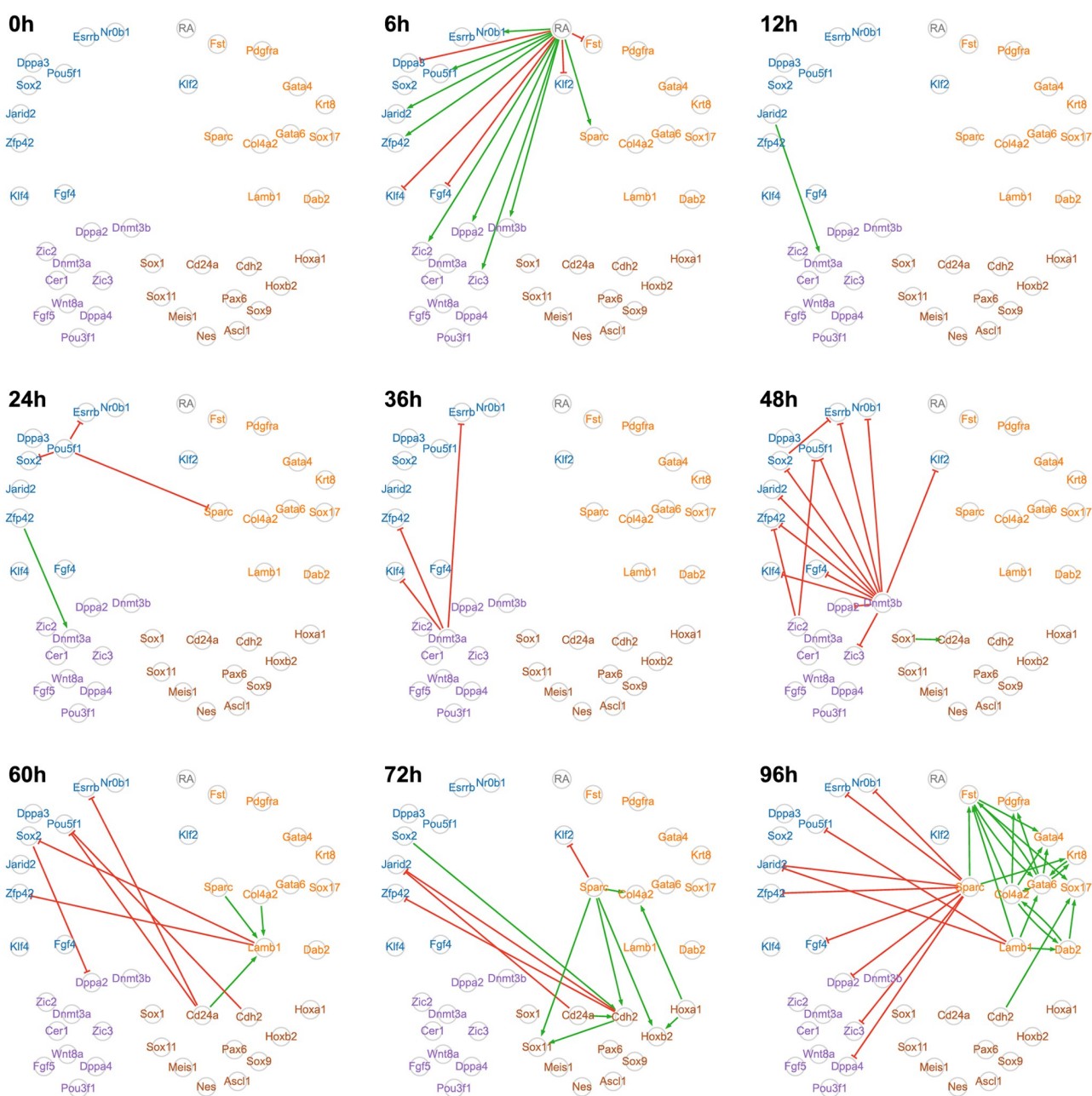

**Fig 6. Time decomposition of the network inferred by CARDAMOM from a real time-stamped scRNA-seq dataset.** Decomposition of the network shown in Fig 5, where each edge appears at the timepoint for which it was detected with the strongest intensity. This dynamic representation highlights a consistent flow of information coming from the stimulus. Gene positions and colors as well as activation and inhibition representations are the same as in Fig 5.

We first decided to verify, as advocated by Soneson et al. [37], the suitability of our generated synthetic data. We used countsimQC [37], a recent tool for comparing multivariate single-cell datasets, already used for benchmarking synthetic scRNA-seq data [38] (see Simulation of the inferred network reproduces the original dataset for more details). The synthetic dataset indeed mimics experimental data for a large number of tested characteristics (S2

Fig). However, we observed that except for correlations, the features considered by count-simQC are also well reproduced by the dataset simulated with the null network. This suggests that countsimQC features are not sufficient for measuring the accuracy of the dataset reproduction.

We then explored the ability of the synthetic dataset to recapitulate more sophisticated dimensions of the experimental data. At that stage, the critical question concerns the temporality of the synthetic data. Indeed, as developed in [15] and [23], none of the algorithms used for the benchmark (and presented in Simulation of the inferred network reproduces the original dataset) allows to take into account the real temporality of the data: the only information they use is the order of the sequence of timepoints at which the cells are measured. It is therefore not necessarily expected that a dataset simulated with the network inferred by CARDAMOM can reproduce the data distribution exactly at the same timepoints. The temporality is taken into account in a second time, by setting the value of the degradation rates from the literature. However, the hypothesis that these degradation rates are not time-dependent (which of course oversimplifies the biological reality [39]) may prevent us from being able to perfectly fit the time-dependent evolution of the data.

We observed that this hypothesis indeed limits our ability to simulate the true dynamics at the last timepoint. In particular, the process seems to accelerate between 72h and 96h and the model cannot be in adequacy with both the dynamics between 0h and 72h, and between 72h and 96h with the same degradation rates. It is important to note that such a global variation in degradation rates has been observed experimentally during the differentiation of chicken erythroid progenitors (see [24] and data at https://osf.io/k2q5b/). We thus decided to multiply the degradation rates by a scaling factor after 72h, allowing the process to reach its final state in time.

We compared these datasets using different metrics. First, we examined the extent to which the simulations matched the experimental marginal distributions of each gene. In Fig 7A, we represent the time-dependent evolution of the p-value of a Kolmogorov-Smirnov test between the GRN-generated distributions and the experimental dataset. We can see that some genes are better fitted than others, as exemplified by the Esrrb gene that is correctly fitted while the Sparc gene seems more difficult to catch (see Fig 7B). We nevertheless observe that for most genes and timepoints, the p-value is above 5%, meaning that the marginal distributions of the experimental data are quite well reproduced by the GRN model. This observation is confirmed by smaller p-values (i.e., significant discrepancies from the experimental data) for many more genes and timepoints when removing interactions between genes (S6 Fig). We compare in Fig 7C the mean Earth Mover Distance (EMD), and in Fig 7D the mean p-value of the Kolmogorov-Smirnov test applied on the 41 genes at each timepoint, between the empirical distributions of the experimental dataset and the two simulated datasets (with and without GRN). We observe that without GRN, the distance between the distributions generated by the model are constantly increasing, that is diverging from the experimental datasets (Fig 7C). This is corroborated by the fact that the mean p-values are decreasing monotonically (i.e. the model's output is more and more significantly different from the experimentally observed distributions, see Fig 7D). The behavior of the GRN-simulated dataset is much closer to the experimental one, as seen from a smaller (and constant) EMD distance as well as a larger mean p-value. However, we observe in Fig 7A that the mid-timepoints (the central portion of the dynamics) seems to be the most difficult to capture, since mid-time p-values are often higher than at the beginning or end of the dynamics. This is corroborated by S3 Fig, where we plotted the temporal marginal distributions of six genes, and where Sox2 and Sparc in particular appear to have a final distribution close to the experimental one but not the correct transient behavior.

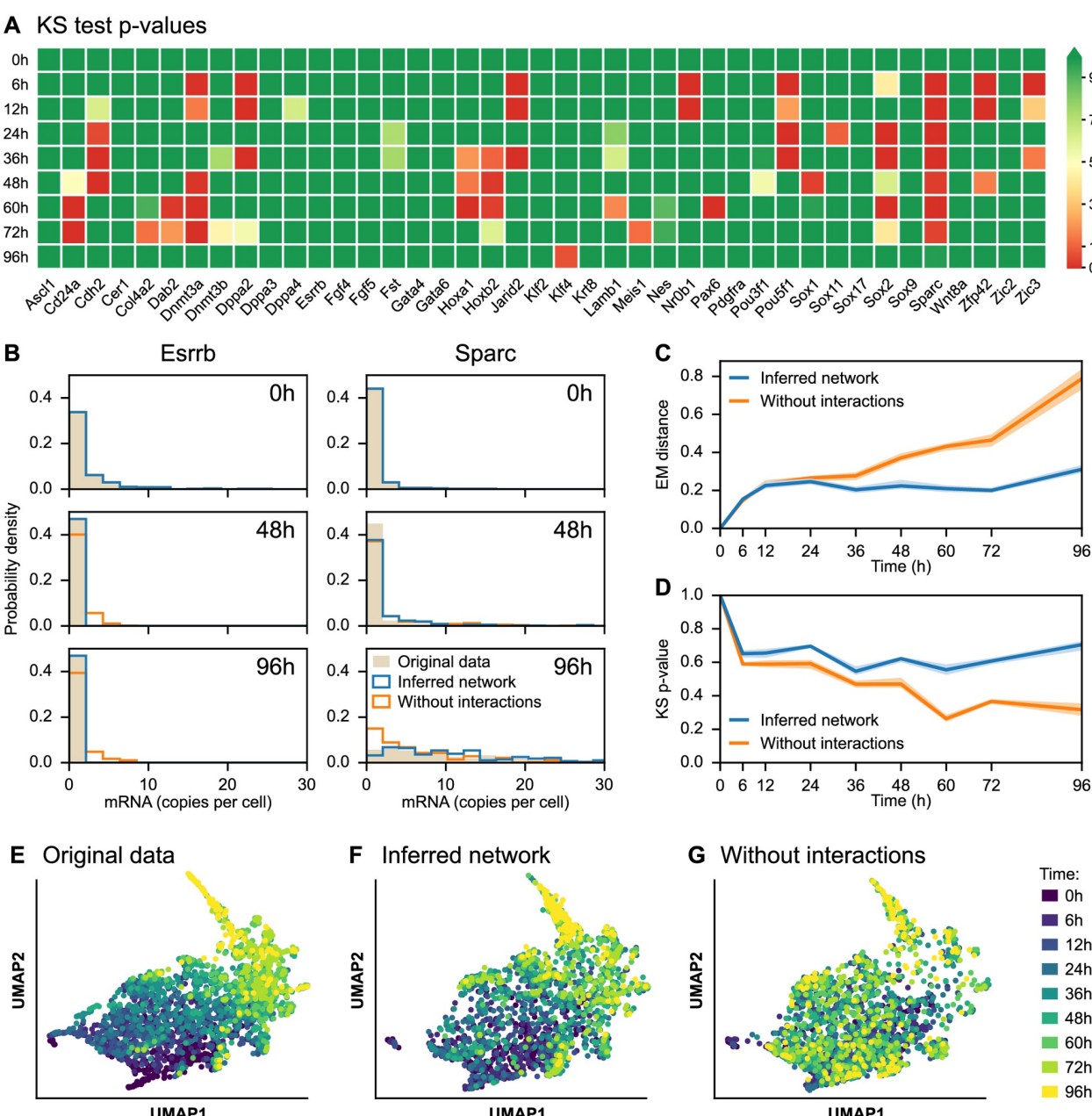

**Fig 7. Inferred network simulations compared to the original dataset.** (A) Heatmap of p-values associated with Kolmogorov–Smirnov (KS) tests between real and simulated mRNA distributions, for each of the 41 genes of the network and for each timepoint. The green color indicates p-values greater than 5%, implying that the model output is not significantly different from the experimental dataset. (B) Time-dependent distributions of Esrrb and Sparc genes for the experimental dataset (*original data*, in beige) and datasets simulated after calibrating the mechanistic model, one including interactions (*inferred network*, in blue) and one obtained after removing interactions (*without interactions*, in orange). (C-D) Average earth's mover (EM) distance (C) and average KS p-value (D) between real and simulated distributions, for the inferred network and without interactions. The dispersion corresponds to the first and ninth decile from ten simulations. (E-F-G) Two-dimensional UMAP representations of the original dataset (E) and the datasets simulated from the inferred network (F) and without interactions (G). In these three plots, Sparc is removed from the genes represented in Fig 5 as its dynamics are not well captured by the mechanistic model, so the three datasets consist of 2449 cells with mRNA levels of 40 genes.

Finally, we were interested in how well we could capture the joint distributions. For this, we compared UMAP representations of the experimental dataset (Fig 7E) and the datasets simulated from the inferred network (Fig 7F) and from the null network (Fig 7G). These three datasets were projected on the same pseudo-axes based on the UMAP computed from the experimental dataset, using the methodology described by McInnes et al. [40]. This common projection allows a better side-by-side comparison between datasets. It is immediately evident that our GRN-generated data points are very closely mimicking the actual experimental data points, and that this resemblance is completely lost if all interactions are removed. The fact that UMAP is not linear requires some precautions, that we discussed in the Simulation of the inferred network reproduces the original dataset section.

To conclude, we observed that the mechanistic model can reproduce the major characteristics of the gene expression patterns observed during a differentiation process examined at the single cell level. It also appears clearly that simulating the model with the network inferred by CARDAMOM significantly improves the fit to the experimental dataset compared to the simulation with the null network (see Fig 7 and S5 Fig).

## Discussion

The major interest of the method we proposed in this work is that it uses the same model for both inferring and simulating a gene regulatory network. The simulation part is not a novelty: a growing number of algorithms are proposed for simulating realistic single-cell gene expression datasets [38], and some have already been used for benchmarking GRN reconstruction methods [16, 17, 41].

Part of the success of our GRN model lies in its ability to reproduce the main characteristics of cell-cell heterogeneity observed in experimental datasets by making stochasticity an inherent part of the model, instead of adding a noise term a posteriori (see Simulation of the inferred network reproduces the original dataset). This is not the case of most algorithms that are used to simulate gene expression data associated to a regulatory network for benchmarking purposes [42, 43], even when they are based on underlying mechanistic models [16]. This point is also crucial for developing analytical results able to be used for the reverse engineering of the model, as it has been done for HARISSA and CARDAMOM.

Among the recently described algorithms, SERGIO [16] and BoolODE [17] are the closest to our work. Nevertheless, we want to highlight some key differences between these two approaches and ours:

1. SERGIO and BoolODE mechanistic models are based on SDEs, treating noise as a Gaussian white noise. This is clearly insufficient to capture the *biological zeros* and Gamma-shaped variability, which in these methods arise only from the addition of technical noise. In our modeling scheme, these features arise naturally from the transcriptional bursting phenomenon.

2. As stressed by the authors, both methods seek to simulate data with an explicit GRN as an input, a forward simulation goal, rather than attempt to estimate it from data, a reverse engineering goal. This is a fundamental difference with our work where we seek *simultaneously* to infer and to simulate data. To the best of our knowledge, the ability to do so using a mechanistic model is a true novelty of our work.

3. Our modeling scheme is amenable to in-depth mathematical analysis [6, 15, 23, 44], which is not the case of SERGIO. For BoolODE, a similar mathematical analysis of the SDE system should be possible, but the fact that gene expression stochasticity is driven only by Gaussian white noise would limit the application of such analysis.

4. The use of a specific module to add technical noise, as well as SERGIO's ability to generate both spliced and unspliced versions of mRNAs are welcome innovation that we will consider in future versions of our work.

We also mention that there has been a recent surge of interest in using generative adversarial networks (GANs) for producing realistic new single-cell transcriptomics data [45]. Although it can be an efficient strategy for data generation and augmentation, its behaves as a black box regarding the underlying biology. Our main added value here is that our model is based on the biophysical reality of the cell and provides a clear materialistic explanation for generated data. Since we have shown that this generative model can be calibrated from single-cell datasets, it can also be used for control purposes, aiming at controlling the cellular phenotype by interfering with the GRN behavior.

While the test case was made using a dataset obtained during a differentiation sequence, one should note that our approach can be applied to any biological process for which time-stamped single-cell transcriptomic data are obtained after applying a given stimulus. When such time-stamped snapshots are not available, the algorithm could in principle take as an input time-reconstructed data (i.e., artificially ordered snapshots). In that case, the quality of the inference will strictly depend on the effectiveness of the time reconstruction algorithm.

Although efficient and promising, the model and the method we presented here have some limitations that are clearly identified, and that should guide future research efforts. First, we consider that the burst frequencies, which are critical parameters of the regulation, are sigmoid functions of protein levels. This implies that the distribution of a gene generally cannot have more than two modes [15], one associated to a low frequency of bursts and one to a high frequency of bursts, which correspond to the regions of the gene expression space where the sigmoid is relatively flat. Thus, if the distribution of a gene is more complex than a mixture of these two modes, the model is not expected to reproduce accurately its dynamics, since minor regulatory interactions might go undetected (especially if a third "hidden" mode is close to one of the two main modes). This seems to be the case for example for the slight decrease of Sparc at t = 24h, which would have been better captured by adding an extra mode. To solve this kind of errors, it may be necessary to complexify the model, for example by modeling the burst rate functions by a multi-layer perceptron rather than a sigmoid as proposed in [15].

We also observe that, as for Sparc, there is a positive correlation between the connectivity of a gene in the network and the complexity of its marginal distribution, leading to lower p-values in Fig 7A for genes that are highly connected. This could be because genes with complex behavior have to be regulated by potentially many other genes for that behavior to be explained, and variations in their expression make them good candidates for explaining variations in the expression of other genes.

Second, CARDAMOM uses the temporality linearly, by taking into account the timepoints one after the other without possibility of backward step. This explains why the algorithm is unable to detect the known fact that a gene like Jarid2 inhibits some Extraembryonic and Neuroectoderm genes: indeed these inhibitions could only be detected by going back in time when the target genes see their expression increase, in order to find the real cause of this effect. Instead, we have seen that the algorithm interprets it as an activation of some other genes. We believe that it could be tackled by taking inspiration from the Recurrent Neural Network (RNN) theory, but it should be achieved while keeping the interpretability of the results. Note that we already developed a similar analogy for the regression step at each timepoint, which can be interpreted as the learning step of a perceptron [15].

From that point of view, if the information was indeed transmitted forward, the network would be supposed to be completed step by step until reaching its complete form. However, two types of incompatibilities may still occur:

1. A direct incompatibility occurs when an edge which has been inferred at a certain time-point is chosen to be reset to a value close to 0 or even to change sign at another one.

2. An indirect incompatibility corresponds to the case where the effect of an edge $(j \rightarrow i)$, which has been inferred at a certain timepoint, is compensated by another edge $(k \rightarrow i)$ at a following timepoint, but that the gene $k$ expression products were high enough at the previous timepoint to thwart the effect of the relation $(j \rightarrow i)$.

This explains why, for the experimental dataset presented in the section, the model is not able to reproduce the behavior of some genes. One good example is Pou5f1 between t = 0h and t = 24h: indeed, the edges that could generate the slight increase of Pou5f1 at t = 6h are thwarted by the edges inferred at the following timepoints which lead to the strong decrease of Pou5f1 after t = 48h. However, these incompatibilities could also have a biological meaning, and be impossible to solve by modifying the regression problems. To go further, it is necessary to discuss the notions of structure and states of the network, which is related to the importance of possibly hidden variables. If the network incorporates all critical nodes, then the structure should not change. But it is different if there are some hidden variables, like genes the level of which are not measured, which results in modifying the network structure. For example, the problem of Pou5f1 that we have presented above could be explained in the following way: at t = 0h, an hidden variable may act on the interaction by preventing its possibility, before the hidden variable disappear at t = 24h. So if the hidden variable was integrated, then the network structure would not change anymore, but it is likely that we should in our case consider a modification of the network structure at t = 24h.

Third, the model does not allow the synthesis and degradation rates to vary over time. We have already mentioned that this was a problem for simulating the passage from t = 72h to t = 96h, and we decided to speed up the last time step for the model to reach its stationary distribution at 96h. We believe that most of the errors observed in the simulation with respect to the experimental dataset could be solved by finding an appropriate degradation rate. Thus, a significant improvement would consist in taking into account at each regression step the size of the time interval, and not only its order in the series of timepoints as it is now the case. This could allow to find a most-likely GRN in accordance to the degradation rate at each timepoint, or even to infer a most-likely degradation rate for each timepoint. However, while the latter case may provide new information on the variation of degradation rates as well as better accuracy on the relative importance of interactions in the GRN, it could also accentuate the problems of identifiability, and should therefore be studied carefully.

Fourth, the model does not take into account proliferation nor apoptosis while studying the stochasticity of the differentiation processes, nor the regulation of the proliferation rate by gene expression products. When sampling a distribution of $n$ cells at a time $t$, the initial condition is built by sampling $n$ cells under the uniform distribution among the set of cells at 0h, and to simulate its evolution during $t$ hours. However, if some cells are supposed to have a higher death rate, and others to have a higher division rate, the process should evolve preferably in a certain direction in the gene expression space, which is going to be ignored in the current version of CARDAMOM. Taking into account these characteristics is a notoriously difficult task: a significant improvement has been recently achieved with Waddington-OT [27, 46], where a stochastic diffusion process models gene expression dynamics. Extending this kind of approach for the mechanistic model will be the subject of future works.

Finally, future versions of our method may consider additional biological features such as spatial cell-cell communication as the advent of multiomics datasets should provide data allowing to analyze the effect of these processes on differentiation, which is not possible in the case of scRNA-seq data without seriously compromising identifiability. We believe that the work presented here could serve as a basis for developing multiscale approaches to differentiation processes.

## Methods

### Mechanistic model of gene regulatory networks

The model used throughout this article is based on a hybrid version of the well-established two-state model of gene expression [6], where a gene is described by the state of a promoter, which can be either *on* or *off*. If the promoter is *on*, mRNAs are being transcribed at a rate $s_0$, which are then translated into proteins at a rate $s_1$. Degradation of both mRNAs and proteins occurs at a rate $d_0$ and $d_1$, respectively. The transitions between the on and off states occur at times of rates $k_{\text{on}}$ and $k_{\text{off}}$ ($k_{\text{on}} \ll k_{\text{off}}$), corresponding to short active periods with high transcription rates, as experimentally observed [47–50]. In this regime, mRNA is then transcribed by bursts of tens to hundreds of molecules. The random times at which these bursts occur are still described by an exponential distribution of parameter $k_{\text{on}}$, and their random size by an exponential distribution with mean $s_0/k_{\text{off}}$. This model is compatible with experimental single-cell data, as steady-state mRNA levels follow for each gene a Gamma distribution, in line with continuous single-cell data [10].

The key idea is to incorporate this model into a network: the burst rate for each gene $i$ is given by a gene-specific function $k_{\text{on},i}^\theta(P)$, where $P$ is the vector of protein quantities (S1 Fig). This function depends on proteins through a GRN, represented by an $n$-by-$n$ matrix $\theta = (\theta_{ij})$ where $n$ is the number of genes in the network. The value of $k_{\text{on},i}^\theta(P)$ then corresponds to the transcriptional burst frequency of gene $i$ given protein levels $P$. Each parameter $\theta_{ij}$ encodes the interaction $j \to i$ with its direction, sign, and intensity. Recent work suggests that burst sizes are smaller and more uniform than previously anticipated [49] therefore leaving more room for burst frequency modulation [51] as a mechanism for gene expression regulation. We therefore consider that interactions come mainly from the modulation of burst frequencies $k_{\text{on},i}^\theta$ and that for any gene $i$, the rates $k_{\text{off},i}$ do not depend on $P$. The burst frequencies can be represented by sigmoid functions [23] as a simplification of the mechanistic form used in [6, 14]:

$$k_{\text{on},i}^\theta(P) = k_{0,i} + (k_{1,i} - k_{0,i})\left(1 + \exp\left(-\beta_i - \sum_{j=1}^{n} \theta_{ij}P_j\right)\right)^{-1}$$

where $k_{0,i}$ (resp. $k_{1,i}$) is the minimal (resp. maximal) burst frequency of gene $i$ and $\beta_i$ is the basal activity of gene $i$, which can be also considered as the constant activity of a set of genes that are not measured but act on the network.

### Simulation of time-stamped datasets

In order to simulate the mechanistic model, we used the simulation module of the HARISSA package [23]. One computational advantage of this method, which consists in sampling burst times with maximum rate and then deciding with an appropriate rule which ones to keep, is that it is guaranteed to be exact without requiring any numerical integration.

To simulate discrete "count" data that are produced by current scRNA-seq technologies, each mRNA level is generated by sampling from a Poisson distribution whose mean is the

simulated expression level. An important mathematical observation is that the resulting cell profiles are then exactly (resp. approximately) distributed according to the discrete-valued "Gillespie" version of the mechanistic model in the absence (resp. presence) of interactions between genes [52].

In order to reproduce *in vitro* experiments for a specific GRN, we use the following method: (1) let the model run for t < 0h until its (stochastic) steady state is reached; (2) introduce at t = 0h a virtual *stimulus* gene with a constant maximal value for its protein. Such stimulus represents a perturbation in the environment of the cells, inducing them to evolve towards a new (stochastic) steady state. For example, in the case of mouse ES cell differentiation, this corresponds to the addition of all-trans RA in the medium. A time-stamped dataset corresponds to the sampling of independent cells at a specific sequence of timepoints (therefore "killing" sampled cells at each timepoint) starting from t = 0h. Namely, for the benchmark of Fig 2, the sequence of 10 timepoints was set to 0, 6, 12, 24, 36, 48, 60, 72, 84, and 96h.

The GRN model parameters ($k_{0,i}$, $k_{1,i}$, $s_{0,i}/k_{\mathrm{off},i}$, $\beta_i$ and $\theta_{ij}$) as well as the degradation rates used for simulating the datasets of Fig 1 can be found online with the code of the CARDA-MOM method. For every gene $i$, we set $k_{0,i} = 0$, $k_{1,i} = 2$ and $s_{0,i}/k_{\mathrm{off},i} = 50$.

## Relevance to biological data

The exact probability distribution associated to the mechanistic model remains unknown for general networks. However, the analysis developed in [15] suggests that the marginal on mRNAs of the distribution at each time $t$ can be reasonably approximated by a Gamma mixture:

$$M_t \sim \sum_{z \in Z} \mu_t(z) \prod_{i=1}^{n} \mathrm{Gamma}\left(\frac{k_{z,i}}{d_{0,i}}, \frac{k_{\mathrm{off},i}}{s_{0,i}}\right), \tag{1}$$

where $Z$ denotes the set of cell types seen as the basins of attraction of the GRN model, $k_{z,i}$ denotes the mode of burst frequency associated to gene $i$ within basin $z \in Z$, and $\mu_t$ is a probability vector describing the relative weight of the basins in the process at time $t$.

The Poissonian layer transforms the Gamma distributions into negative binomial (NB) distributions, which gives:

$$M_t \sim \sum_{z \in Z} \mu_t(z) \prod_{i=1}^{n} \mathrm{NB}\left(\frac{k_{z,i}}{d_{0,i}}, \frac{k_{\mathrm{off},i}}{k_{\mathrm{off},i} + s_{0,i}}\right). \tag{2}$$

Such mixture distributions are known to be compatible with discrete single-cell data [1, 11]. In particular, we recover the second order relationship between pairs of variables that are characteristic of experimental datasets. Indeed, we remark that the mean of a negative binomial distribution $\mathrm{NB}(a, b)$ is $m = \frac{a(1-b)}{b}$ and its variance $v = \frac{a(1-b)}{b^2}$, which implies that:

$$CV^2 = \frac{v}{m^2} = \frac{1}{a(1-b)} = \frac{1}{b}\frac{1}{m}.$$

Thus, for every gene $i$, by replacing $b$ by $\frac{k_{\mathrm{off},i}}{k_{\mathrm{off},i}+s_{0,i}}$, we see that the relation $CV^2 \sim \frac{1}{m}$, which is characteristic of cell-cell heterogeneity in single-cell data, is well verified by the mechanistic model provided that $k_{\mathrm{off},i}$ does not depend on protein levels. This also argues in favor of the assumption that a GRN does not affect significantly the burst size. Note however that following this criterion, any model generating negative binomial distributions could be considered as realistic. Such criterion is therefore not sufficient for characterizing the accuracy of gene expression

models, especially when the simulated distributions arise from a phenomenological "ad hoc" noise term added to fit experimental datasets [16].

## Tested algorithms

The six algorithms used for the benchmark represent together the main categories of GRN inference methods presented in [7]:

- GENIE3 [18], which computes the regulatory network for each gene independently, using tree-based ensemble methods to predict the expression profile of each target gene from the profiles of all the other genes;

- PIDC [19], which infers an undirected network using the notion of mutual information;

- SINCERITIES [20], which uses Granger causality after computing temporal changes in gene expression through the distance between two consecutive timepoints of the marginal distributions;

- SCRIBE [21], which is based on the notion of conditioned Restricted Directed Information and ideally needs real cell trajectories, which is unrealistic experimentally. We then pseudo-temporally order the time-stamped synthetic data used for the benchmark with two methods, one using a pseudotime algorithm and the other using an optimal coupling method with optimal transport, following the idea developed in [27]. We also tested real trajectories in order to compare the performances. The results of this algorithm are presented separately, due to the difference in the information that is needed.

- HARISSA [6, 23] and CARDAMOM [15], which are based on the mechanistic model presented above. Both algorithms reconstruct the network by solving a set of regression problems, based on two distinct mathematical analyses of the same model: HARISSA solves a maximum likelihood problem for the protein distributions after estimating a most-probable position for the protein levels in each cell; CARDAMOM compares the function $k_{\mathrm{on}}$ to the modes of a joint mRNA distribution previously inferred, in a two-step procedure. Although there are few differences from previous publications regarding these tools, they have been slightly improved here by taking into account the advantages of each, to make them more efficient and compatible with each other. The differences are described in the file `cardamom_vignette.pdf` on the associated Git repository.

## Measuring algorithm performance for the benchmark

We evaluated the GRN inference algorithms on simulated datasets using the area under the precision-recall curve (AUPR). Since inferring these coefficients is a notoriously difficult task [17], we do not take into account diagonal coefficients of the GRN matrix, which correspond to self-regulations. Interestingly, for the datasets used in the benchmark (see Fig 1), the self-regulation of a gene is generally well detected by CARDAMOM at a significantly higher level than for genes without self-regulation. However, the values inferred without self-regulation remain high relative to the other inferred interactions, which pulls down the AUPR scores while in practice not affecting the associated GRN dynamics. Since this effect of diagonal coefficients tends to lower the AUPR scores for all methods in a similar way, the choice not to take them into account is well justified. Note also that we chose precision-recall (PR) curves rather than receiver operating characteristic (ROC) curves because of the well-known class imbalance problem. Indeed, the sparsity hypothesis suggests that the number of interactions expected for a network of size $n$ is smaller than half of the total number of possible interactions ($n^2$): it is

then natural to focus on minimizing false positives (interactions that are detected but not present) rather than false negatives (interactions that are present but not detected), which explains the preference of PR over ROC.

## Experimental dataset

We used data collected from a differentiation experiment of mouse embryonic stem cells induced by all-trans retinoic acid treatment [22]. This scRNA-seq dataset consists of 9 timepoints (0, 6, 12, 24, 36, 48, 60, 72, and 96h), each timepoint containing between 137 and 335 sampled cells after pre-processing (272 on average, for a total number of 2449 cells). To limit artificial correlations between genes (due to a multiplicative cell-specific technical factor mainly related to the reverse-transcription step), we selected cells with a total number of UMI counts $\geq$ 2000 in line with Semrau et al. [22], which resulted in keeping only 2449 out of 3456 measured cells.

On the other hand, we did not normalize cells by their respective total UMI counts and argue that this type of normalization is hazardous in the case of single-cell data. Indeed, such "library sizes" are small compared to bulk data (because 1 sample = 1 cell instead of many cells) and are in fact biologically fluctuating, likely reflecting the transcriptional bursting phenomenon (this is easily seen when simulating "perfect" data from the mechanistic model, see S2 Fig). In practice, since the CARDAMOM inference method starts with a binarization step (applying a specific, statistically derived threshold to each gene based on the mechanistic model), a multiplicative factor on each cell should not have too much impact as long as the number of cells is large enough. More generally, we argue that such normalization of cells should rather be "soft-coded" as a random factor to be estimated within a statistical framework.

The total number of genes measured in this experiment is 17452, which is much larger than in our benchmark. As they are unlikely to all be important in characterizing the differentiation process, we decided to restrict our analysis to a panel of 41 genes that had previously been identified as key marker genes for pluripotency, post-implantation epiblast, neuroectoderm and extraembryonic endoderm [22]. This number of genes allows to infer a network rich enough to make cell types emerge in a non-trivial way, while keeping a reasonable statistical power regarding the number of sampled cells. Note that the speed of the algorithms (S1 Table) would allow a much larger subset of genes to be used: the limiting factor here is not computational speed but statistical power resulting from the number of cells available (see Fig 2).

## Calibration of the mechanistic model

The principle of CARDAMOM is based on a two-step procedure:

- In a first step, we find the set of parameters $\alpha$ defining the mixture of negative binomial distributions (2) that best fits the data. The only parameter that can vary over time is the mixture proportion parameter $\mu_t$ (allowing to estimate, for each gene $i$, the mean burst size $s_{0,i}/k_{\text{off},i}$ and the values $k_{0,i}$ and $k_{1,i}$ of the typical modes associated to the function $k_{\text{on},i}$). Note that all model parameters are estimated except the degradation rates, which are constant for each gene and scale the dynamics of gene expression. See [15, Appendix A] for a full description of these parameters and their precise links with $\alpha$.

- In a second step, we calibrate the basal activity and interaction parameters $\beta_i$ and $\theta_{ij}$ in order to approximate this mixture distribution. The interaction parameters $\theta_{ij}$ are then updated at each timepoint sequentially to match the mixture parameters.

The degradation rates are set to values found in tables from the literature [36]. Since many genes do not appear in these tables, we decide to set the same value for all the genes belonging to the same functional group identified in [22] (see S2 Table). Details concerning the two steps can be found in [15]. Note that the first step has been modified since the original publication in order to replace the MCMC algorithm, which was used to find the parameters of the negative binomial distributions associated to each cell type, by a faster variational method.

As mentioned previously, the model cannot be in adequacy with both the dynamics from 0h to 72h and from 72h to 96h with the same degradation rates for the experimental dataset used in, due to the acceleration of the process between 72h and 96h. For this reason, we decided for our simulation to multiply the synthesis and degradation rates between the last two time points (equivalent to the multiplication of the last timepoint) by a factor $f = 6$, large enough to reach the steady state between 72h and 96h. This factor was found empirically as the minimum integer factor such that the stationary distribution is reached at 96h: thus any factor greater than 6 would lead to the same results.

## Comparing datasets using countsimQC

We used countsimQC [37] to compare the experimental dataset, the dataset simulated with the inferred network and the dataset simulated with the null network (S2 Fig). Although there are no significant difference between the first two datasets, we observe in S2F Fig, that the correlations between genes are not perfectly reproduced (they are clearly more accurate than for the dataset simulated with the null network). This gap between the correlations between genes is also illustrated in S4 Fig, which compares joint distributions between the simulated dataset (with the inferred network) and the experimental one for three pairs of genes at the final timepoint. We observe that if the global form of the correlation is respected, they are not as strong in the simulated dataset as in the experimental dataset. This suggests that the inferred GRN recovers the true correlations but not with the right intensity, which may be due to the sensitivity of the model to the value of its parameters.

The fact that except for the correlations (sample-sample and feature-feature), the statistical characteristics explored by countsimQC are also well reproduced by the dataset simulated with the null network, suggests that they are generally not sufficient for measuring the accuracy of a dataset reproduction. This is partly due to the fact that any calibration of the model with the right scaling parameters but not the right GRN should match most of these characteristics. In that meaning, the successes of simulation algorithms prior to our work are limited when measured with similar criteria. Our methodology, for which we used distinct criteria which are particularly well illustrated in S5 Fig and Fig 7, then appears as a significant improvement in the field of executable GRN inference.

## Comparing datasets using UMAP

Since UMAP is not linear, the projections of datasets shown in S5 Fig are likely to force the projected data to be artificially close to the reference dataset. Thus, we decide to present two figures similar to Fig 7E, 7F and 7G, but where instead of projecting the simulated dataset on the pseudo-axis corresponding to the projection of the experimental dataset, we project both datasets together and show separately the cells corresponding to each dataset. Using this methodology, we represented in S5A Fig the UMAP projection of the experimental dataset and in S5B Fig the one of the dataset simulated with the inferred network. We did the same for the experimental dataset (S5C Fig) and the dataset simulated with the null network (S5D Fig). Then, although they have different representations, Figures S5A and S5C Fig represent the same dataset, and the difference comes from the second dataset (the simulated one) with

which the reduction has been performed. This allows to emphasize that the representation of a distribution of cells with UMAP is very sensitive to the choice of the data that are integrated in the projection. Once again, we observed that the dataset simulated using the inferred network does seem much closer to the experimental dataset than the one simulated with the null network: in particular, S5B Fig demonstrates that the UMAP projection of the dataset with network is close to the one of the experimental dataset both in the arrangement of the cells between the different timepoints and in the general form of the subspace occupied by the cells, which is not the case for the UMAP projection of the dataset simulated with the null network, represented in S5D Fig.

## Supporting information

**S1 Table. Runtime of inference methods.** Related to Fig 2. Average runtime for inferring a network from datasets simulated with tree-like networks for which the results of the inference are represented in Fig 2, for the six algorithms that are used in the benchmark. Timings measured on a 16-GB RAM, 2.4 GHz Intel Core i5 computer.
(CSV)

**S2 Table. Degradation rates.** Related to Fig 7. Numerical values of mRNA and protein degradation rates (in $h^{-1}$) used for data simulation (sim.), compared with experimental (exp.) measures from the literature. Group abbreviations: Pluri = Pluripotency, Epi = Post-implantation epiblast, Neuro = Neuroectoderm, Endo = Extraembryonic endoderm.
(CSV)

**S1 Fig. Mechanistic GRN model.** Related to Fig 1. Graphical (**A**) and mathematical (**B**) descriptions of the mechanistic model for the dynamics of a gene $i$. (**C**) Bursts of mRNA occur at random times with rate $k_{on,i}$ and their size follows an exponential distribution $\mathcal{E}(k_{off,i}/s_{0,i})$. The variables $M_i$ and $P_i$ describe respectively the mRNA and protein quantities associated to gene $i$ in the cell. The vector of protein levels is denoted by $P = (P_1, \cdots, P_n)$ while $\theta$ denotes the GRN which couples the genes together through functions $k_{on,i}$.
(PDF)

**S2 Fig. Statistical characteristics.** Related to Fig 7. Comparison of various statistical characteristics across datasets using the countsimQC package. Each plot shows the experimental dataset (*real*, in blue) and datasets simulated from the mechanistic model calibrated by CARDAMOM, including interactions (*network*, in green) and without interactions (*naive*, in red). Each dataset consists of 41 genes (*features*) measured in 2433 single cells (*samples*). (**A**) Distribution of "library sizes", defined as the total read count in each sample. (**B**) Association between the library size and the fraction of zeros observed per cell. (**C**) Distribution of the fraction of zeros observed per cell. (**D**) Distribution of cell-cell correlations, based on random cell pairings. (**E**) Distribution of the fraction of zeros observed per gene. (**F**) Distribution of gene-gene correlations, based on all possible gene pairings. Notably, the cell-cell correlation (**D**) bimodal pattern shows two possible pairings of cells: pairs with similar expression profiles (same genes *on*, same genes *off*) and therefore positively correlated, and pairs with opposite, "antinomic" profiles and therefore negatively correlated. This pattern is an indirect sign of the emergence of different cell types, a characteristic that is clearly not reproduced in the absence of interactions between genes (naive dataset).
(PDF)

**S3 Fig. Marginal distributions.** Related to Fig 7. Comparison between empirical distributions along timepoints, for six genes that have been found to play a key role in the regulation of the

process (as visible in Fig 5). The experimental dataset (in beige), the dataset simulated from the inferred network (in blue) and the dataset simulated without interactions (in orange) correspond to Fig 7E, 7F and 7G, respectively.
(PDF)

**S4 Fig. Joint distributions.** Related to Fig 7. Comparison of the joint distributions of three pairs of genes at the final timepoint between the experimental dataset (**A**) compared to the dataset simulated when the mechanistic model is calibrated by CARDAMOM (**B**) and the dataset simulated without interactions (**C**). The genes in each pair are expected to have interactions (direct or indirect) in the network represented in Fig 5.
(PDF)

**S5 Fig. UMAP representations.** Related to Fig 7. Two-dimensional UMAP representations of the experimental dataset (*original data*) and datasets simulated after calibrating the mechanistic model with CARDAMOM, one including interactions (*inferred network*) and one obtained after setting $\theta = 0$ (*without interactions*). The four plots are based on two different projections, computed after merging the experimental dataset with (**A-B**) the simulated dataset including interactions or (**C-D**) the simulated dataset without interactions. Hence **A** and **C** are two representations of the exact same data, while **B** is to be compared with **A**, and **D** is to be compared with **C**.
(PDF)

**S6 Fig. KS tests for the null network.** Related to Fig 7. Heatmap of p-values associated with Kolmogorov–Smirnov (KS) tests between real mRNA distributions and the ones simulated from the null network (without interactions), for each of the 41 genes of the network and for each timepoint. The green color indicates p-values greater than 5%, implying that the model output is not significantly different from the experimental dataset.
(PDF)

**S7 Fig. Network inferred by SINCERITIES from the real time-stamped scRNA-seq dataset.** Related to Fig 5. The SINCERITIES inference method was applied to the experimental dataset from [22] restricted to a panel of 41 marker genes identified by the authors. The network structure is obtained by keeping only the 5% strongest activations (green arrows) and inhibitions (red blunt arrows) acting on each gene. Genes are colored according to four groups related to different cell states (pluripotency, post-implantation epiblast, neuroectoderm, extraembryonic endoderm) following the proposed classification of [22]. Edges supported by a ChIP-seq interaction are marked with black dots (see main text for the definition of what is considered as an interaction) and edges that are not supported are marked with white dots: this concerns only the edges starting from the RA stimulus, Pou5f1, Sox2, and Jarid2. Edges for which we have no reliable information have no mark.
(PDF)

## Acknowledgments

We would like to thank especially Christophe Arpin, Thomas Lepoutre, Anton Crombach and Arnaud Bonnaffoux for critical reading of the manuscript. We also thank all members of the SBDM and Dracula teams, and of the SingleStatOmics project, for providing such stimulating working environment. We finally thank the BioSyL Federation, the LabEx Ecofect (ANR-11-LABX-0048) and the LabEx Milyon of the University of Lyon for inspiring scientific events.

## Author Contributions

**Conceptualization:** Elias Ventre, Ulysse Herbach, Thibault Espinasse, Olivier Gandrillon.

**Formal analysis:** Elias Ventre, Ulysse Herbach, Gérard Benoit.

**Funding acquisition:** Olivier Gandrillon.

**Investigation:** Elias Ventre, Ulysse Herbach.

**Methodology:** Elias Ventre, Ulysse Herbach, Thibault Espinasse.

**Software:** Elias Ventre, Ulysse Herbach.

**Supervision:** Thibault Espinasse, Olivier Gandrillon.

**Validation:** Elias Ventre, Ulysse Herbach, Thibault Espinasse, Gérard Benoit, Olivier Gandrillon.

**Visualization:** Elias Ventre, Ulysse Herbach, Olivier Gandrillon.

**Writing – original draft:** Elias Ventre, Gérard Benoit, Olivier Gandrillon.

**Writing – review & editing:** Elias Ventre, Ulysse Herbach, Thibault Espinasse, Gérard Benoit, Olivier Gandrillon.

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
