## [Decision Letter · Decision Letter 0]

28 Nov 2022

Dear %TITLE% Ventre,

Thank you very much for submitting your manuscript "One model fits all: combining inference and simulation of gene regulatory networks" for consideration at PLOS Computational Biology. As with all papers reviewed by the journal, your manuscript was reviewed by members of the editorial board and by several independent reviewers. The reviewers appreciated the attention to an important topic. Based on the reviews, we are likely to accept this manuscript for publication, providing that you modify the manuscript according to the review recommendations.

You will see that the reviewers have appreciated the topic and the writing of the manuscript. However, some of the reviewers ask for more details on specific computational details, and suggest some clarifications in the explanation of the methods.

Sincerely,

Carl Herrmann, Ph.D.

Academic Editor

PLOS Computational Biology

Jason Haugh

Section Editor

PLOS Computational Biology

Reviewer's Responses to Questions

**Comments to the Authors:**

Reviewer #1: The authors present a computational technique for gene regulatory network (GRN)-based simulation of single-cell transcriptomics data sets, with the underlying GRN being reconstructed from a real data set. This functionality is not present in existing GRN-based simulation tools.

The tool called HARISSA, published previously, is a very carefully designed mechanism-based simulator of single-cell expression dynamics under the influence of a GRN. The tool called CARDAMON, also previously published, is capable of inferring the GRN underlying a single cell transcriptomic data set, under the same mechanistic model used by HARISSA.

Evaluation of GRN inference is performed first on synthetic data generated by HARISSA. In these evaluations, the CARDAMOM algorithm previously published by the authors performs best among several appropriate competing methods. This demonstrates that the CARDAMOM method is indeed able to reconstruct GRNs when the synthetic data arise from its underlying model, as expected.

The authors then apply their methods to partially reconstructing the GRN for Retinoic Acid-induced differentiation of mouse ESCs and validate it based on ChIP data, again noting improvements over appropriate baseline (existing) methods. This is a useful exercise for a future user of the tool, even though the actual performance metrics (AUPRC, AUROC ~ 0.6 compared to ~0.5 random baseline) are not stellar.

The last section is a very interesting and exciting aspect of the work, where networks inferred from real data are then used to simulate synthetic data which is shown to match the real data. This is a clear advance in the field.

Overall, this study is a rigorous and innovative next step at the forefront of scRNA-seq data modeling and simulation, and to some extent of GRN reconstruction as well. I wholeheartedly recommend publication of the work.

Reviewer #2: In the manuscript the authors combine two computational tools for the simulation and inference of GRNs and test them in multiple scenarios including in silico and in vitro data. The results are analyzed with rigorous statistical tools and are well organized.

I appreciate the effort that the authors put on the analysis specially comparing with other existent tools, I think that this is crucial for good research on this topic were new algorithms are appearing continuously. While I am familiar with the simulation and inference of GRN, I am not familiar with the state of the art of libraries applied to scRNAseq data and therefore it is hard for me to identify the novelty of the manuscript. Both tools used in this manuscript HARISSA and CARDAMOM have been developed in the past. I would have appreciated to have a more explicit introduction to the technicalities of the different tools used and a more clear identification of the differences to previous publications on these two tools.

Reviewer #3: This well-written manuscript presents a compelling new approach to predict gene regulatory networks (GRNs) by combining two previously-published methods HARISSA and CARDAMOM. The key component of HARISSA that is of use here is a GRN simulation algorithm based on a "multi-agent" generalization of a standard two-state stochastic, mechanistic model of gene expression. This model captures bursting transcriptional dynamics. CARDAMOM is a scalable algorithm for inference of the parameters of this model, and hence, the underlying GRN itself. In combination, these methods permit both inference of a GRN and its simulation.

The analysis is very thorough. The choice of algorithms for comparison is very good. The authors first focus on small, synthetic, networks with well-understood dynamics. HARISSA and CARDAMOM clearly outperform other algorithms and when they do not, it is clear that even a simple correlation-based algorithm can perform well. The decision to use precision-recall curves here is cogent. I liked the care given to SCRIBE, the scalability analysis for trees, and the thorough analysis of the limit of performance with respect to the number of cells and timepoints.

The analysis of the experimental data is also thorough and well-considered. The performance (AUPRC) of HARISSA and CARDAMOM drops to around 0.6 but is still larger than the other algorithms. The highlight of this manuscript is the simulation of the inferred network and the comparison of the resulting data to the original measurements. This analysis, which is rarely seen in GRN inference papers (mainly because it is simply not possible), will make this manuscript stand out when published.

The interpretation of results is very careful. The authors

I have a few suggestions for further strengthening the manuscript.

1. The authors should clarify whether they used every pair of genes not present in the ground truth as a negative example to compute the number of false positive and true negatives. In this context, I am puzzled as to how the random classifier can have a precision of 0.47 in Figure 4B. This high value suggests a ground truth network where nearly half the possible edges are present.

2. The authors should include a supplement that studies the network inferred by GENIE3 (or SINCERITIES) in a manner similar to page 11 and Figure 5.

3. In Figure 7A, the p-values should be corrected for testing multiple hypotheses. I am not sure if they are. The authors could reduce the number of tests by performing one K-S test for each gene considering its expression over all time points. It will also be instructive to see such a plot for the naive model without interactions.

4. On page 3, lines 79-80, the authors state that "existing GRN-based simulation tools, which are generally based on more phenomenological than mechanistic models". It may be that my understanding of "phenomenological" and "mechanistic" is different, but isn't BoolODE based on simulating a Boolean (mechanistic) model rather than a phenomenological one? In Discussion, the authors state that SERGIO uses a mechanistic model.

MINOR COMMENTS

page 18, line 447: "by going up the arrow of time" may be better phrased as "by going back in time".

page 21, line 558: Change "bursts" to "burst".

page 21, line 561: The meaning of the phrase "a synthetic noise well adapted" is unclear. Perhaps this sentence should be rephrased.

Several citations are missing page numbers.

Reviewer #4: Review in attachment

**Have the authors made all data and (if applicable) computational code underlying the findings in their manuscript fully available?**

Reviewer #1: Yes

Reviewer #2: Yes

Reviewer #3: **No: **The software for both HARISSA and CARDAMOM are available but I do not see the network files for the toy networks in Figure 1, the simulated data for the corresponding analyses, or the ground truth networks for Figure 4. The authors should provide code to generate all the results in their manuscript, even if in the form of Jupyter notebooks.

Reviewer #4: Yes

PLOS authors have the option to publish the peer review history of their article (what does this mean?). If published, this will include your full peer review and any attached files.

Reviewer #1: No

Reviewer #2: No

Reviewer #3: No

Reviewer #4: No

Figure Files:

Data Requirements:

Reproducibility:

References:

---

## [Decision Letter · Decision Letter 1]

17 Feb 2023

Dear %TITLE% Ventre,

We are pleased to inform you that your manuscript 'One model fits all: combining inference and simulation of gene regulatory networks' has been provisionally accepted for publication in PLOS Computational Biology.

Best regards,

Carl Herrmann, Ph.D.

Academic Editor

PLOS Computational Biology

Jason Haugh

Section Editor

PLOS Computational Biology

Reviewer's Responses to Questions

**Comments to the Authors:**

Reviewer #3: The revision is very good. The authors have addressed my concerns well.

Reviewer #4: The authors have clearly answered the remarks and questions and have amended the manuscript which has clarified the text

**Have the authors made all data and (if applicable) computational code underlying the findings in their manuscript fully available?**

Reviewer #3: Yes

Reviewer #4: Yes

PLOS authors have the option to publish the peer review history of their article (what does this mean?). If published, this will include your full peer review and any attached files.

Reviewer #3: No

Reviewer #4: No

---

## [Editor Report · Acceptance letter]

22 Mar 2023

PCOMPBIOL-D-22-01102R1 

One model fits all: combining inference and simulation of gene regulatory networks

Dear Dr Gandrillon,

I am pleased to inform you that your manuscript has been formally accepted for publication in PLOS Computational Biology. Your manuscript is now with our production department and you will be notified of the publication date in due course.

With kind regards,

Zsofia Freund
